# The Role and Mechanism of Polysaccharides in Anti-Aging

**DOI:** 10.3390/nu14245330

**Published:** 2022-12-15

**Authors:** Xinlu Guo, Junjie Luo, Jingyi Qi, Xiya Zhao, Peng An, Yongting Luo, Guisheng Wang

**Affiliations:** 1Department of Nutrition and Health, China Agricultural University, Beijing 100193, China; 2Department of Radiology, the Third Medical Centre, Chinese PLA General Hospital, Beijing 100039, China

**Keywords:** polysaccharide, anti-aging, mechanism, *Caenorhabditis elegans*, *Drosophila melanogaster*, mice, aging indicators

## Abstract

The elderly proportion of the population is gradually increasing, which poses a great burden to society, the economy, and the medical field. Aging is a physiological process involving multiple organs and numerous reactions, and therefore it is not easily explained or defined. At present, a growing number of studies are focused on the mechanisms of aging and potential strategies to delay aging. Some clinical drugs have been demonstrated to have anti-aging effects; however, many still have deficits with respect to safety and long-term use. Polysaccharides are natural and efficient biological macromolecules that act as antioxidants, anti-inflammatories, and immune regulators. Not surprisingly, these molecules have recently gained attention for their potential use in anti-aging therapies. In fact, multiple polysaccharides have been found to have excellent anti-aging effects in different animal models including *Caenorhabditis elegans*, *Drosophila melanogaster,* and mice. The anti-aging qualities of polysaccharides have been linked to several mechanisms, such as improved antioxidant capacity, regulation of age-related gene expression, and improved immune function. Here, we summarize the current findings from research related to anti-aging polysaccharides based on various models, with a focus on the main anti-aging mechanisms of oxidative damage, age-related genes and pathways, immune modulation, and telomere attrition. This review aims to provide a reference for further research on anti-aging polysaccharides.

## 1. Introduction

Aging is a complex natural phenomenon, which is manifested as structural and functional degeneration. It is the inevitable result of the synthesis of a number of processes, influenced by many physiological and psychological factors such as heredity, immunity, and environment [1]. In recent years, with economic and medical advances, the world’s life expectancy reached 71 in 2021 [2]. However, the elderly proportion of the population continues to grow. Worldwide, the proportion of people over 60 years old in the total population has increased from 9.9% in 2000 to 13.7% in 2021, with Eastern and Southeast Asia reaching 17.4%, and Europe and North America reaching 25.1% [2]. It is estimated that by 2050 there will be more than 1.5 billion people over the age of 65 [2]. Aging is usually accompanied by cardiovascular and cerebrovascular diseases, diabetes, and other chronic diseases, and it is a great burden to the economy, health care, and society [3]. According to published data, issues surrounding the aging population are becoming increasingly more serious and will continue to worsen unless action is taken [2]. To increase healthy lifespan, one strategy is to create anti-aging medications.

Aging is accompanied by a variety of physiological processes, such as a decline in immunity, a slowing of basal metabolism, and a reduction in activity of antioxidant-related enzymes [4]. Therefore, potential anti-aging drugs can be screened from substances that have inhibitory effects on these processes. According to the different modes of action, many anti-aging theories have been proposed, including programmed theories and damage theories [5]. Programmed theories involve deliberate deterioration with age, and are likely the most relevant to aging genes as well as the endocrine system. Some scientists believe that aging is not programmed, but rather involves the accumulation of damage such as oxidative damage, mitochondrial DNA damage, and genome damage. These two theories have been thoroughly explored in previous publications [5]. Current anti-aging drugs, such as rapamycin and metformin, have demonstrated not only strong anti-aging effects in a variety of model organisms through different ways such as mTOR and AMPK, but also demonstrated promising results in preliminary clinical trials [6,7]. Zhang et al. reviewed studies on the retardation of aging by rapamycin in a variety of animal models and organ systems [8]. Rapamycin plays an anti-aging role mainly by inhibiting mTOR, which is important for mitochondrial function, metabolism, and maintenance of stem cells. According to the literature, metformin delays aging through various mechanisms, such as regulating the synthesis and degradation of age-related proteins, maintaining telomere length, and reducing DNA damage [9]. However, due to the lack of large-scale and long-term clinical trials, the above drugs still have some deficits related to safety and long-term use. For example, rapamycin has been demonstrated to extend the lifespan of mice, but it did not improve age-related characteristics; furthermore, rapamycin may cause thrombocytopenia, nephrotoxicity, and other side effects [10]. Long-term use of metformin may lead to vitamin B12 deficiency, lactic acid accumulation, and even lactic acid poisoning [11]. Taken together, the current research on anti-aging drugs is still in a relatively preliminary stage. It is also necessary to develop naturally effective anti-aging drugs with fewer side effects, thus allowing for long term use.

At present, a number of studies have demonstrated that polysaccharides extracted from natural resources have a wide range of pharmacological effects, such as anti-inflammatory, anti-oxidative, and immune modulatory effects [12]. Moreover, polysaccharides may have unique advantages in terms of side effects and long-term use due to their low cytotoxicity [12], and are expected to contribute to the development of novel anti-aging drugs or supplements [12]. Notably, some polysaccharides have been utilized in clinical practice. For example, heparin is used as an anticoagulant drug [13]. *Poria cocos* polysaccharide oral liquid has been approved for the treatment of many diseases, such as cancer and hepatitis [14]. In terms of anti-aging, although there is no polysaccharide drug that can be applied in clinical practice, many polysaccharides have demonstrated good effects in various animal models such as *Caenorhabditis elegans* (*C. elegans*), *Drosophila melanogaster (D. melanogaster)*, and mice. For instance, angelica sinensis polysaccharide (ASP) and astragalus polysaccharide (APS) have been demonstrated to significantly prolong the life of *C. elegans* and *D. melanogaster*, and also have protective effects on the liver, kidney, brain, and other important organs in mice, thus demonstrating great potential for anti-aging therapy [15,16,17,18,19,20]. This article reviews the progress of polysaccharides in the field of anti-aging based on different animal models and relevant mechanisms, and aims to support the development of novel polysaccharide anti-aging drugs.

## 2. Research on Anti-Aging Polysaccharides

Currently, a number of studies have demonstrated that polysaccharides exert good anti-aging effects. Of course, individual polysaccharides may have different anti-aging effects and mechanisms across animal models. Accordingly, each model has its own advantages and indicators of aging. Although the effects observed in animal models are not always comparable to those observed in humans, these studies function to demonstrate the potential of polysaccharides for use in anti-aging therapy. In the literature, the most common animal models are *C. elegans*, *D. melanogaster*, and mice, which we briefly analyze here based on their merits and limitations. Furthermore, this section highlights the common anti-aging indicators and polysaccharide studies based on these common animal models.

### 2.1. C. elegans and D. melanogaster

*C. elegans* and *D. melanogaster* are common anti-aging model organisms with similar advantages. *C. elegans* organisms undergo some age-related changes in the reproductive and nervous systems. As they grow older, they become dull and their physiological functions decline, making them easy to observe and operate [21]. Secondly, they have a short life cycle and life span. The average life span in *C. elegans* is about 20 days and, therefore, the full life cycle can be observed in a relatively short time [21]. Finally, a portion of *C. elegans* genes is highly conserved with humans, which is beneficial to the study of the molecular mechanism of aging [22]. Similar to *C. elegans*, *D. melanogaster* also has the advantages of rapid reproduction, short life cycle (about 3 months), easy operation, and gene conservation, and these model organisms are also used often in anti-aging research [22,23].

The common aging indicators of *C. elegans* and *D. melanogaster* are shown in Table 1 and Table 2, respectively. Among them, the most intuitive index is their life span under normal conditions or under various stress conditions (such as H_2_O_2_ and heat), which can be used to evaluate the effect of polysaccharides on life span. It is also an essential index to study the anti-aging effect. However, this index is only applicable to models with short life cycles such as *C. elegans* and *Drosophila* and, therefore, has limitations [24].

Other indicators used to characterize the senescence of *C. elegans* and *Drosophila* can be divided into two main approaches. First, free radicals and antioxidant-related indicators can be detected to determine whether polysaccharides delay aging by affecting oxidative stress. This involves the content or activity of common antioxidant enzymes and metabolites in the body, including superoxide dismutase (SOD), catalase (CAT), glutathione peroxidase (GSH-Px), malondialdehyde (MDA), and reactive oxygen species (ROS) [25,26]. ROS are produced by the normal metabolism of living organisms as well as in response to environmental stimuli. ROS include a variety of chemical substances, including superoxide anions and hydroxyl radicals. In general, higher metabolic rate contributes to production of more ROS, and ultimately correlates to a shorter lifespan [25]. In some ways, the appropriate amount of ROS also plays a fundamental role in maintaining homeostasis. However, excessive ROS will lead to REDOX imbalance and DNA damage, especially mitochondrial DNA damage. Damaged mitochondria will release more ROS, leading to further damage, which becomes a vicious cycle [25]. Naturally, the body also has antioxidant defenses against ROS damage, such as SOD, CAT, and GSH-Px [26]. Therefore, by increasing the expression and activity of antioxidant enzymes, oxidative stress damage can be efficiently controlled, thereby alleviating some effects on aging. The second approach involves the study of the genes and pathways related to aging to clarify their mechanism of action. The specific mechanisms are introduced in the third section of this review.

Previous studies have demonstrated that many types of polysaccharides, including Lycium barbarum polysaccharides (LBP), Bletilla striata polysaccharide (BSP), and Polysaccharides from Rehmannia glutinous (PRG), delay the senescence of *C. elegans* by affecting gene expression in the insulin/insulin-like growth factor signaling (IIS) pathway (Table 1) [22,27,28]. For example, Zhang et al. compared the effects of LBP on the life span of *sir2.1*, *daf-12* and *daf-16* mutants, respectively, and found that after knocking out these three genes, LBP had a smaller effect on life span than in wild type organisms [27]. However, after using LBP, the *sir2.1* mutant with *daf-16* knockout demonstrated no significant difference in the life span compared with the wild type, and the *daf-12* mutant with *daf-16* knockout did not extend life span. These results indicate that LBP affects the senescence of *C. elegans* by regulating *daf-12* and *daf-16* genes. Furthermore, *sir2.1* may be dependent upon *daf-16* to play an anti-aging role. In addition to the IIS pathway, it has been proposed that the regulation of *atf-6* by *miR-124* may be another pathway through which polysaccharides regulate the life span of *C. elegans* [15]. As shown in Table 2, similar results have been observed in *Drosophila*. LBP has also been observed to delay aging and improve stress resistance in *Drosophila*, although its effect may be achieved through the expression of longevity genes such as *MTH*, *Hep*, *Rpn11*, as well as age-related signaling pathways such as MAPK, mTOR, and S6K; this is slightly different from that in *C. elegans* [29]. Overall, these polysaccharides demonstrate good anti-aging effects in both models.

### 2.2. Mice

Mice are one of the ideal animal models for studying human disease. The normal life span of a mouse is about 3 years, and it takes time and effort to directly detect the life span of individual mice. Therefore, mice with accelerated aging or shortened life span are frequently employed to research aging. The most popular of these is D-galactose (D-Gal)-induced aging mice. D-Gal is a reducing sugar. At higher doses, it is converted to aldose and hydroperoxides, leading to ROS accumulation and accelerated aging [33].

There are many aging indicators in hematopoietic stem cells (HSCs) as well as in livers, kidneys, and brains of mice. Aging can be characterized by evaluating aging-related genes and antioxidant indicators in various organs of mice. However, different organs also have specific biochemical markers. For HSCs, senescence can be characterized by cell cycle, viability detection, and senescent cell staining [17]. For organs such as liver, kidney, and brain, aging can be observed through histopathological analysis. Moreover, serum can be separated to detect the content of serum markers, such as the liver markers aspartate aminotransferase (AST) and alanine aminotransferase (ALT), as well as the kidney markers blood urea nitrogen (BUN) and creatinine (CRE) [18].

There have been many anti-aging studies of polysaccharides in mice, particularly in many plant polysaccharides, such as APS and ASP [15,16,17,18,19,20]. They have been demonstrated to play a role in delaying aging in multiple organs of mice, and the possible anti-aging mechanisms have been explored. As shown in Table 3, ASP is crucial for delaying aging in HSCs, liver, kidney, and brain, and this involves several underlying mechanisms. First, APS has strong antioxidant and anti-inflammatory activity, which increases the activity of SOD, CAT, GSH-Px, and other enzymes, reduces the content of ROS and MDA, and inhibits D-Gal-mediated inflammatory factors, including iNOS and COX-2, so as to inhibit oxidative stress and inflammation [18]. Second, APS works by targeting the age-related genes and pathways, including the p16^INK4a^-Rb pathway and the p19^Arf^-Mdm2-p53-p21^CIP1/Waf^ pathway [34]. P16^INK4a^ is a tumor suppressor that is expressed in most senescent cells. It activates the tumor suppressor pRB in some cells, causing chromatin damage and silencing of pro-proliferative genes [35]. The p53 pathway has similar effects [34]. Moreover, p53 can also act through the Wnt/β-catenin pathway. Joerg et al. discussed its role in a variety of model organisms, demonstrating that the Wnt pathway plays a vital role in organ development, stem cell differentiation, and other processes [36]. Notably, the excessive activation of this pathway leads to varying degrees of stem cell senescence. ASP has been demonstrated to reduce the mRNA and protein expression of these pathways, thereby delaying aging [34].

In addition to plant polysaccharides, the effects of microbial polysaccharides on aging have been studied. For example, Zhang. et al. have investigated that intracellular zinc polysaccharides from Grifola Frondosa SH-05 have significant antioxidant and anti-aging effects through detection of antioxidant activity in multiple organs [37]. These findings provide ideas for the development of new anti-aging agents.
nutrients-14-05330-t003_Table 3Table 3Anti-aging study of polysaccharides using mice as a model.PolysaccharidesMain Aging IndicatorsMechanismReferenceASPCell analysis (cell cycle and the propotion of senescent cell); age-related genes (*p53)*; Telomere and telomerase ASP can antagonize X-ray-induced senescence of HSC, possibly by affecting telomere and *p53* expression[17]ASPAntioxidant indexes (SOD, GSH-Px, MDA, AGEs); liver tissue markers (ALT, AST, TBil, histomorphology)ASP can antagonize D-Gal induced liver injury in aging mice, possibly by inhibiting oxidative stress[38]ASPAntioxidant indexes (SOD, GSH-Px, MDA, AGEs); DNA damage markers (8-OH-DG); renal tissue markers (BUN, Crea, UA, Cysc, histomorphology)ASP can antagonize D-Gal induced subacute kidney injury in mice, possibly by inhibiting oxidative stress injury[39]ASPDNA damage markers (ROS, 8-OHdG, 4-HNE and γ-H2A.X); age-related pathways (P16^Ink4a^-Rb, p19^Arf^-Mdm2-p53-p21^CIP1/Waf^ and Wnt/β-catenin) ASP has antioxidant ability but the effect is not as good as V_E_; ASP delays senescence by affecting the expression of senescence signaling pathway factors[34]ASPAge antioxidant indexes (SOD, CAT, GSH-Px); organ indexes; immune modulatory (inflammatory factors)ASP effectively protects liver and kidney from D-Gal-induced injury in mice, which may be related to the reduction of oxidative response and inflammatory stress[18]ASPCell analysis (cell proliferation and the propotion of senescent cell); antioxidant indexes (SOD, MDA, T-AOC, ROS); age-related genes (*P53, P21*); immune modulatory (inflammatory factors)ASP may delay brain aging in mice by regulating the number and function of hippocampal neural stem cells, reducing the oxidative damage, inhibiting the expression of inflammatory cytokines and aging genes[19]AcAPS ^7^ and its major purified fractions (AcAPS-1, AcAPS-2 and AcAPS-3)Liver and kidney tissues damage markers (AST, ALT, ALP, BUN, CRE, ALB, histomorphology); antioxidant indexes (SOD, CAT, GSH-Px, MDA)AcAPS-2 has a good protective effect on liver and kidney, among which rhamnose and glucose play a more important role[40]IZPS ^8^Antioxidant indexes (SOD, MDA, T-AOC); brain tissue damage markers (histomorphology)IZPS can increase the antioxidant activity[37]MWP ^9^Antioxidant index (SOD, CAT, GSH-Px, MDA); neuronal apoptosis MWP can improve antioxidant ability and inhibit neuronal apoptosis[41]APSAntioxidant indexes (SOD, CAT, GSH-Px, MDA, ROS); mitochondrial damage markers (permeability)APS can improve antioxidant capacity, inhibit mitochondrial damage and swelling[20]APSCell analysis (propotion of senescent cell); age-related genes (*P16, P21, P53*); mitochondrial damage marker (NCLX, ATP, cytochrome C oxidase activity and the oxygen consumption rate); immune modulatory (inflammatory factors)APS can regulate the senescence of vascular endothelial cells induced by high glucose through enhancing the expression of NCLX, inhibiting inflammasome activation, improving mitochondrial dysfunction and promoting autophagy[42]YLSP ^10^Antioxidant indexes (SOD, CAT, GSH-Px, MDA and AGEs); immune modulatory (cytokine levels, organ indexes); aging-related genes (*P21, P53*)YLSP may inhibit the aging process by enhancing antioxidant activity and immune function and regulating the expression of aging-related genes[43]^7^ Acidic-extractable polysaccharides of Agaricus bisporus (AcAPS) includes Fuc, Rha, Xyl, Gal, Glu and Man. ^8^ Intracellular zinc polysaccharides from Grifola frondosa SH-05 (IZPS) mainly consist of Rha, Ino, and Glu. ^9^ Polysaccharide from Malus micromalus Makino fruit wine (MWP). ^10^ Yulangsan polysaccharides (YLSP).


### 2.3. Cell Lines Studies

At present, there are few clinical studies on polysaccharides. Furthermore, aging studies evaluating polysaccharides and common aging markers have largely relied on the use of human cell lines (Table 4) and not human tissues, and thus the actual effect of polysaccharides on the human body remains unknown. Due to the need for long-term clinical trials to ensure the safety and efficacy of human anti-aging drugs, current anti-aging drugs remain lacking in this aspect of testing. However, as discussed above, it can be observed that plant polysaccharides represented by ASP, APS, and LBP have demonstrated excellent anti-aging effects in various models [15,16,17,18,19,20,29], demonstrating the potential of polysaccharides as anti-aging drugs. However, these findings require long-term clinical verification.

## 3. Anti-Aging Mechanism of Polysaccharides

Typical features of aging include a decline in basal metabolism accompanied by the decreased immune function and antioxidant capacity. The studies mentioned above suggest that polysaccharides may act to delay aging by regulating metabolism and immunity. Here, we introduce four widely accepted mechanisms to provide references for subsequent research on the anti-aging mechanisms of polysaccharides.

### 3.1. Oxidative Damage

Living organisms produce free radicals during normal physiological activities, especially from the mitochondrial electron transport chain. At the same time, there are also free radical scavenging systems in the body, such as SOD, CAT, and GSH-Px [26]. However, with increased age, the balance between the two is difficult to maintain, resulting in an excess of free radicals [25]. Unsaturated fatty acids in biofilm are easily converted to lipid peroxides by free radicals. These products cause damage to proteins, nucleic acids, and other substances, which accelerates aging [47]. Additionally, since mitochondria provide energy for cells and regulate the cell cycle, excess free radicals can cause serious damage to mitochondria and accelerate the process of aging [48]. As shown in Figure 1, polysaccharides act to up-regulate the expression of antioxidant-related enzyme genes through the nuclear factor-E2-related factor 2-antioxidant response element (Nrf2-ARE) pathway, so as to remove excess free radicals and achieve the purpose of anti-aging. Nrf2 is normally bound to Keap1 and is then rapidly degraded, resulting in a low level in cells. Under the influence of polysaccharides, Nrf2 enters the nucleus and interacts with the ARE to increase the expression of antioxidant genes [47]. For example, Yang et al. isolated polysaccharides from fruit wine and identified their scavenging effects on free radicals and anti-aging effects in vivo [41]. Zhu et al. found that CCP significantly prolonged the lifespan of *Drosophila* by increasing the activity of SOD, CAT, and GSH-Px [32]. Finally, Li et al. demonstrated that APS acts to protect mitochondria, likely by scavenging ROS and increasing the activities of antioxidant enzymes [20]. These results demonstrate that polysaccharides not only act to improve the activity of antioxidant enzymes, but also inhibit cell apoptosis and the formation of lipid peroxides.

### 3.2. Age-Related Genes and Pathways

Specific genes can affect the lifespan of an organism. At present, studies have identified parts of genes (e.g., *p53* and *p21*) and pathways (e.g., IIS pathways and Wnt/β-catenin pathways) that are associated with aging and longevity, as discussed in the second section of this review [43]. Mutations or changes in the expression of these genes can significantly impact lifespan. Under normal circumstances, the p53 protein as a tumor suppressor is swiftly ubiquitinated by MDM2 and subsequently targeted for degradation by the ubiquitin-proteasome system. In conditions of DNA damage, p53 is activated by posttranslational modifications, which inhibit the interaction of p53 with MDM2 and lead to the accumulation of p53 [49]. However, p53 not only triggers apoptosis, but also induces cell cycle arrest. Therefore, the expression of the *p53* gene is closely related to cell aging and DNA damage [49]. Another protein, p21, is involved in many important cellular processes, including apoptosis and DNA replication [50]. It can be used as a cell cycle regulatory protein to inhibit the activities of various cell cycle-dependent kinases, thus promoting cell cycle arrest [50]. Additionally, these two proteins are important factors in the p19^Arf^-Mdm2-p53-p21^CIP1/Waf^ pathway, as discussed in Section 2.2. It has been reported that polysaccharides can affect the cell cycle by regulating the expression of these genes. For example, Yulangsan polysaccharide may enhance antioxidant activity and immune function by regulating expression of the age-related genes *p53* and *p21* [43].

Besides these genes, IIS is a well-studied pathway that is highly conserved in various organisms such as *C. elegans*, *Drosophila*, and mice [51]. This pathway is related to growth, development, reproduction, and aging. Under normal circumstances, IIS is active and is mainly involved in the phosphorylation of DAF-2, AGE-1, and other kinases, ultimately affecting the activity of DAF-16 transcription factors [51]. After phosphorylation, DAF-16 interacts with the 14-3-3 protein and is anchored in the cytoplasm. Inhibition of the IIS pathway by external intervention reduces the phosphorylation of DAF-16, facilitating translocation to the nucleus [52]. Furthermore, protein phosphatase 4 complex promotes the recruitment of RNA polymerase to assist the DAF-16 gene in activating the transcription of stress resistance and longevity-promoting genes, ultimately prolonging life [52]. At present, several studies have demonstrated that many kinds of polysaccharides act to prolong the lifespan in *C. elegans* and *Drosophila* models [16,22,28].

As mentioned in Section 2.2, the Wnt/β-catenin pathway is one of the most important pathways related to developmental processes, and excessive activation of Wnt/β-catenin pathway may lead to varying degrees of stem cell senescence [36]. This pathway will only be activated when the Wnt protein binds to the Frizzled receptor and co-receptor LPR-5/6. Once this tri-molecular complex is formed, Dishevelled will be recruited. DVL is then phosphorylated, resulting in the inhibition of GSK-3β and accumulation of free β-catenin. The free β-catenin then translocates to the nucleus and activates genes that influence cellular processes [53,54].

### 3.3. Immune Modulation

The immune function of the body declines with increased age, and this mainly manifests as an infective response to infectious disease and a decreased response to vaccines [55]. Polysaccharides have been demonstrated to improve immune function by regulating inflammatory factors as well as affecting immune cells and immune organs, so as to achieve the purpose of anti-aging [43,56].

At the molecular level, various inflammatory cytokines and chemokines, such as interleukin-1β (IL-1β) and tumor necrosis factors (TNFs), show increased gene and protein expression in senescent cells compared to non-senescent cells [57]. In addition, senescent cells show excessive activation of a variety of inflammatory mediators. This overactivation of inflammatory cytokines associated with aging may be realized through the P38 and NF-ĸB pathways [57]. Mo et al. investigated whether ASP could protect against the D-Gal-induced aging in mice by attenuating inflammatory responses [18]. The results indicated that ASP could indeed inhibit the expression of inflammatory factors related to the NF-ĸB pathway, such as iNOs and COX-2, and the infiltration of the hepatic leucocytes in aged mice. This example demonstrates that polysaccharides can delay aging by attenuating inflammation. Notably, the accumulation of advanced glycosylation end products (AGEs) is an aging indicator [58]. Glycosylation begins with the carbonyl group of a carbohydrate and the amino group of a protein, which reacts to form a Schiff base. Schiff bases, however, are unstable and undergo a series of reactions, such as rearrangements, which eventually turn into AGEs. Then, AGEs trigger ROS overload and stimulate proinflammatory cytokine synthesis and release [58,59]. Therefore, the levels of AGEs and inflammatory cytokines that affect T-cell immune responses serve as important indicators of aging [60].

At the cellular level, senescent cells with complex senescence-related secretory phenotypes (SASPs) are beneficial to tissue repair and slow down the development of cancer in the short term. However, these cells may aggravate a variety of diseases in the long term [61]. T cells, macrophages, NK cells, and other immune cells may be recruited by SASP factors to eliminate senescent cells and maintain homeostasis [61]. Polysaccharides bind to specific receptors on the surface of immune cells; for instance, ganoderma lucidum polysaccharides bind to dectin-1, mannose receptor, and Toll-like receptor 4 to activate the immune response [62]. However, the mechanism of the interaction between various polysaccharides and different immune cells requires further study.

At the organ level, aging is accompanied by the degeneration of immune organs, such as the thymus and spleen. Therefore, changes in immune organs can characterize aging to a certain extent. According to the histopathological analysis of mouse liver tissue, Xia et al. found that mice without ASP had serious liver injury along with degenerative changes in hepatocytes, decreased hepatic glycogen, and accumulated AGEs [38]. In contrast, mice treated with ASP had more glycogen as well as less liver damage and AGEs.

### 3.4. Telomere Attrition

Telomeres maintain the integrity and stability of chromosome structure and telomerase endows cells with the ability of self-renewal [63]. However, as cells continue to divide, telomerase activity is weakened and telomeres are gradually shortened or are completely lost, eventually leading to cell aging [64]. It has been demonstrated that polysaccharides improve telomerase activity and prevent telomere loss, such that cells can divide normally and aging is delayed. For example, ASP increases telomere length and improves telomerase activity to delay senescence of HSCs induced by X-ray [17]. The anti-aging effect of polysaccharides isolated from the roots of Polygala tenuifolia is partly mediated by the down-regulation of Bmi-1 expression and the activity of telomerase in cells [65]. Fucoidan induces apoptosis and inhibits telomerase activity, which may be mediated by the ROS-dependent inactivation of the PI3K/Akt pathway [66].

## 4. Conclusions and Perspective

With the social pension burden constantly increasing, the development of safe and effective anti-aging drugs to prolong health and life is receiving an increasing amount of attention. Polysaccharides, which are natural antioxidants, not only have a wide variety of sources, but also exert anti-inflammatory and anti-aging activities with little side effects on human health. Based on the above advantages, polysaccharides are expected to become novel anti-aging drugs. However, the anti-aging research specific to polysaccharides is still in the preliminary stages, and there are still a lot of issues to be solved.

Most polysaccharides used in the literature are extracted from plants. As such, their components are complex and unclear, and may include components other than pure polysaccharides. Extraction and purification methods have a great influence on the experimental results.The absorption mechanism and anti-aging mechanism of polysaccharides requires further exploration.Aging is a process of many physiological changes, and it involves multiple factors and organs. For animals with short life cycles such as *C. elegans* and *D. melanogaster*, life span can be directly detected. For other organisms with longer life spans, there are fewer intuitive indicators to characterize aging, which is an indirect representation of one or several factors or organs.There remains a lack of long-term and large-scale clinical testing of polysaccharides as potential anti-aging drugs.

Therefore, in the future, it will be of great value to further analyze the composition and function of polysaccharides, and to study their anti-aging mechanisms through clinical research. Polysaccharides may be useful in combination with other drugs to enhance anti-aging effects and reduce unwanted side effects or damage. In addition, with the ongoing expansion of anti-aging research, it is necessary to construct a relatively comprehensive aging evaluation system that is more conducive to the progress of this research.

## Figures and Tables

**Figure 1 nutrients-14-05330-f001:**
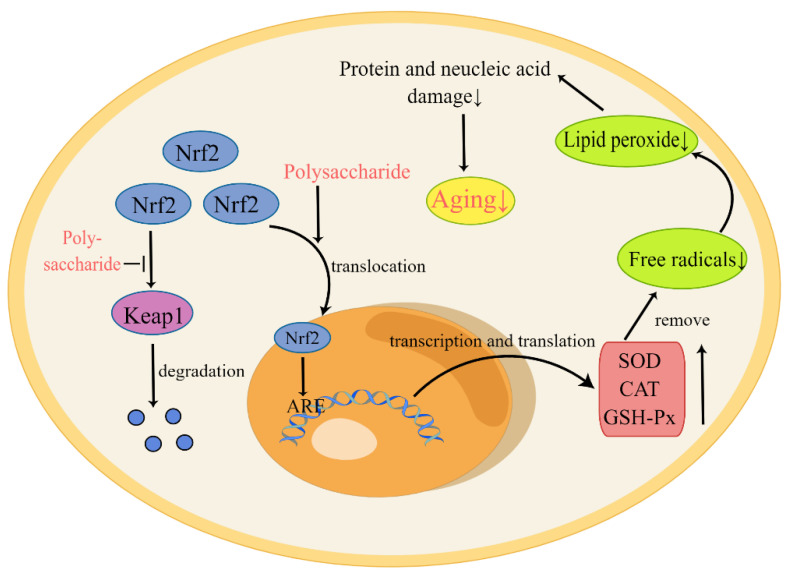
Polysaccharides delay aging by increasing the expression of antioxidant enzymes. In the figure, “↑” indicates elevated substance levels and “↓” indicates the decrease of substance levels. Under normal circumstances, Nrf2 exists in the cytoplasm and is degraded after binding to Keap1, thus maintaining a low level of Nrf2 in the cytoplasm. In the presence of polysaccharides, the binding of Nrf2 to Keap1 is inhibited, and the translocation of Nrf2 is promoted, such that it enters the nucleus and interacts with AREs. The activation of AREs promotes transcription and translation of antioxidant-related genes. These antioxidant enzymes, such as SOD, CAT, and GSH-Px, act to eliminate free radicals and delay aging.

**Table 1 nutrients-14-05330-t001:** Anti-aging study of polysaccharides using *C. elegans* as a model.

Polysaccharides	Main Aging Indicators	Mechanism	Reference
APS	Lifespan; age-related genes (*miR-124* and *atf-6*)	The lifespan of *C. elegans* is prolonged by APS with the regulation *atf-6* by *miR-124*	[15]
LBP ^1^	Lifespan under normal and stress conditions; age-related genes (*sir-2.1*, *daf-12*, and *daf-16*)	The effects of LBP on *C. elegans* health and aging were modulated by *sir-2.1*, *daf-12*, and *daf-16*	[27]
CCP ^2^	Lifespan; age-related genes (HSP)	CCP can protect nerves and delay aging	[30]
BSP ^3^	Lifespan under normal and stress conditions; age-related pathway (IIS pathway)	BSP affects nematode life through the IIS pathway	[22]
PRG ^4^	Aging pigment (lipofuscin); antioxidant enzymes (SOD and CAT and AGEs); age-related pathway (IIS pathway)	PRG can enhance the ability of nematodes to resist oxidative stress and delay senescence through IIS	[28]
LPR ^5^	Lifespan under normal and stress conditions; aging pigment; antioxidant indexes (SOD, CAT, MDA, ROS)	LPR can improve the antioxidant defense system and scavenge free radicals of nematodes to extend the lifespan without toxicity	[26]
Panax notoginseng polysaccharide	Lifespan under normal and heat stress conditions; antioxidant indexes (SOD, CAT, MDA, ROS)	The scavenging ability of it is weak, but it can improve the activity of antioxidant enzymes, reduce the formation of lipid peroxides, and significantly prolong the life span	[31]

^1^ Lycium barbarum polysaccharides (LBP) consisted of mannose, glucose, galactose, protein and uronic acid. ^2^ Coptis chinensis polysaccharide (CCP). ^3^ Bletilla striata polysaccharide (BSP). ^4^ Polysaccharides from Rehmannia glutinous (PRG) mainly consist of galactose, glucose and protein. ^5^ Polysaccharide from roots of Lilium davidii var. unicolor Cotton (LPR).

**Table 2 nutrients-14-05330-t002:** Anti-aging study of polysaccharides using *Drosophila* as a model.

Polysaccharides	Main Aging Indicators	Mechanism	Reference
CP ^6^	Lifespan under normal and oxidative stress conditions; antioxidant indexes (GSH-Px, MDA, *CAT*, *SOD1* and *MTH*)	CP70 can up-regulate the antioxidant related genes *CAT*, *SOD1* and *MTH* to prolong the lifespan of *Drosophila*	[32]
APS	Lifespan under normal and oxidative stress; antioxidant indexes (*Sod1, Sod2, Cat*); age-related pathway (IIS pathway)	APS can extend the lifespan of *Drosophila* by affecting antioxidant capacity and IIS pathway	[16]
LBP	Lifespan under normal and stress conditions; antioxidant indexes (SOD, CAT, MDA); expression of aging-related pathways (MAPK, TOR, S6K) and genes (*Hep, MTH,* and *Rpn11*)	The anti-aging activity of LBP is related to the expression of aging related pathways and longevity genes	[29]

^6^ Cordyceps cicadae polysaccharides (CP).

**Table 4 nutrients-14-05330-t004:** Anti-aging study of polysaccharides in human cell lines.

Polysaccharides	Objects	Main Aging Indicators	Mechannism	Reference
ASP	Homo sapiens bone marrow/stroma cell line	Cell analysis; antioxidant indexes (ROS, SOD, GSH-Px); DNA damage markers (8-OHdG, γH2AX)	ASP protects cells from chemotherapy injury by reducing the oxidative damage and improving hematopoietic function	[44]
Transfersomes containing EGCG and hyaluronic acid	Human keratinocyte cell lines	Cell analysis (viability); antioxidant indexes (ROS and MDA); skin aging genes (*MMP2* and *MMP9*)	Transfersomes have excellent antioxidant ability, inhibit collagen degradation, and enhance cell viability and skin penetration	[45]
TFPS ^11^	Human skin fibroblasts	Cell analysis (viability and apoptosis); ROS; aging-related genes (*p16, p21, p53, SIRT-1*)	TFPS attenuates oxidative stress and apoptosis induced by hydrogen peroxide in skin fibroblasts by upregulating *Sirt1* expression	[46]

^11^ Tremella fuciformis polysaccharide (TFPS).

## Data Availability

Not applicable.

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
