# Peer review of "The Role and Mechanism of Polysaccharides in Anti-Aging"

_nutrients, 2022, doi:10.3390/nu14245330_

Round 1

Reviewer 1 Report

The manuscript by Guo and co-workers reviews scientific evidences concerning the use of several polysaccharides with antiaging effect in three different in vivo experimental models, namely C. elegans, D. melanogaster and mice, and in vitro. Although the topic is of great interest, authors fail in their objective because several important reasons. Mainly, authors do not approach to the study of aging from a deep and formal way. They frequently use through the manuscript informal and imprecise expressions. On the other hand, the depth and extent with which the subject has been addressed seems insufficient.

Here are some specific comments:

 Keywords: Please, refer to specific experimental models investigated in the manuscript.

The way to refer to aging theories in line 47 is quite imprecise. Could authors refer to the different antiaging theories in a more specific way?

Lines 48-49: Authors say: “Current anti-aging drugs, such as rapamycin and metformin, have shown not only strong anti-aging effects through these ways in a variety of model organisms” What ways refer authors to?

Lines 64-64: Authors estate that “Polysaccharides, as natural macromolecules, have strong biological activities and low cytotoxicity” What kind of strong biological activities do the authors refers to? Is a intrinsic question to have biological activities because are natural macromolecules?

Line 67: “A lot of studies” is not a scientific way to refer to a number of studies.

Line 69. Please change “anti-oxidation” by “anti-oxidative” and “immune regulatory” by immune-modulatory”.

Lines 88-89. This sentence has grammar mistakes and also a poor description of these experimental models.

Lines 95-96. What does means that “reproductive and nervous systems are intact”?

Line 98: “Capable of rapid reproduction”?

Table 1: Authors should include abbreviations to all molecules described (e.g. APS). Please, use a foot note to describe these abbreviations. The same for Tables 2, 3 and 4.

Table 1: when referring to LBP authors use “Atheticism”.

Tables:  Overall, there is a very poor and unprecise way to refer to aging biomarkers.

Liune 158: “Mice are physiologically similar to humans”

Line 200. Please, refer to “Cell lines studies” instead to “Human”

Line 211: Authors write “Anti-aging mechanisms” and then in point 3.1, 3.2, 3.3 and 3.4 name antiaging theories. This reviewer believe that this is not the proper way to describe molecular mechanisms. Instead, authors should describe particular mechanisms instead to refer to some types of aging theories which, by the way, they do not refer precisely and correctly, with the exception of the free radical theory. 

Lines 217 to 237. After a not orthodox description of the free radical theory of aging, authors only mention one single reference concerning polysaccharides and these aspects.

Figure 1 title: the tithe of figure 1 does has no sense.

Line 3.2: Gene or genetic theory does not exist as it in terms of aging theories. On the contrary, there is a group of gene-related theories. This paragraph has no relationship with such genetic theories as it.

Author Response

Comments for Transmission to Authors

The manuscript by Guo and co-workers reviews scientific evidences concerning the use of several polysaccharides with antiaging effect in three different in vivo experimental models, namely C. elegans, D. melanogaster and mice, and in vitro. Although the topic is of great interest, authors fail in their objective because several important reasons. Mainly, authors do not approach to the study of aging from a deep and formal way. They frequently use through the manuscript informal and imprecise expressions. On the other hand, the depth and extent with which the subject has been addressed seems insufficient.

RESPONSE: We sincerely thank the reviewer for the encouraging and insightful comments. Our manuscript has undergone extensive English revisions, and we have made corresponding changes based on the valuable suggestions of the reviewer to improve our manuscript.

Here are some specific comments:

Keywords: Please, refer to specific experimental models investigated in the manuscript.

RESPONSE: We thank the reviewer for this suggestion, and we have changed the keyword “animal model” to specific experimental models “C. elegans; Drosophila melanogaster; Mice”.

The way to refer to aging theories in line 47 is quite imprecise. Could authors refer to the different antiaging theories in a more specific way?

RESPONSE: Thank you for pointing out this, and we have referred to the different antiaging theories in a more specific way.

Lines 48-49: Authors say: “Current anti-aging drugs, such as rapamycin and metformin, have shown not only strong anti-aging effects through these ways in a variety of model organisms” What ways refer authors to?

RESPONSE: Thank you and we have revised.

Lines 64-64: Authors state that “Polysaccharides, as natural macromolecules, have strong biological activities and low cytotoxicity” What kind of strong biological activities do the authors refers to? Is a intrinsic question to have biological activities because are natural macromolecules? 

RESPONSE: Thank you for pointing out this. We have revised and introduced the biological activities in a more specific way. As for the second question, we have changed the description more clearly in revised manuscript.

Line 67: “A lot of studies” is not a scientific way to refer to a number of studies.

RESPONSE:  Thank you for this suggestion and we have revised.

Line 69. Please change “anti-oxidation” by “anti-oxidative” and “immune regulatory” by immune-modulatory”.

RESPONSE:  Thank you for this suggestion and we have revised.

Lines 88-89. This sentence has grammar mistakes and also a poor description of these experimental models.

RESPONSE:  Thanks and we have revised.

Lines 95-96. What does means that “reproductive and nervous systems are intact”?

RESPONSE:  Thank you and we have changed the description more clearly.

Line 98: “Capable of rapid reproduction”?

RESPONSE:  Thanks and we have changed this description.

Table 1: Authors should include abbreviations to all molecules described (e.g. APS). Please, use a foot note to describe these abbreviations. The same for Tables 2, 3 and 4.

RESPONSE: Thank you for your suggestion and we have changed.

Table 1: when referring to LBP authors use “Atheticism”.

RESPONSE: Thanks and we have deleted this unimportant aging indicator.

Tables:  Overall, there is a very poor and unprecise way to refer to aging biomarkers.

RESPONSE: Thank you for pointing out this and we have changed the way that we refer to aging biomarkers

Line 158: “Mice are physiologically similar to humans”

RESPONSE: Thanks and we have deleted this sentence.

Line 200. Please, refer to “Cell lines studies” instead to “Human”

RESPONSE: Thank you for this suggestion and we have changed.

Line 211: Authors write “Anti-aging mechanisms” and then in point 3.1, 3.2, 3.3 and 3.4 name antiaging theories. This reviewer believe that this is not the proper way to describe molecular mechanisms. Instead, authors should describe particular mechanisms instead to refer to some types of aging theories which, by the way, they do not refer precisely and correctly, with the exception of the free radical theory.

RESPONSE: Thank you for pointing out these issues and giving suggests. We have changed the description more precisely and correctly in revised manuscript.

Lines 217 to 237. After a not orthodox description of the free radical theory of aging, authors only mention one single reference concerning polysaccharides and these aspects.

RESPONSE: Thank you for this suggestion and we have added more references in this part.

Figure 1 title: the title of figure 1 does has no sense.

RESPONSE: Thank you for pointing out this and we have changed the title of figure 1.

Line 3.2: Gene or genetic theory does not exist as it in terms of aging theories. On the contrary, there is a group of gene-related theories. This paragraph has no relationship with such genetic theories as it.

RESPONSE: Thank you and we have changed the way we describe these mechanisms.

Reviewer 2 Report

In this review the authors describe anti-aging effects of polysaccharides isolated from different plant material. The review provides a nice overview and I only have a few recommendations. The authors should mention what sugars are in these mixes (which has been published) and the English is sometimes a bit awkward.

Author Response

Comments for Transmission to Authors

In this review the authors describe anti-aging effects of polysaccharides isolated from different plant material. The review provides a nice overview and I only have a few recommendations. The authors should mention what sugars are in these mixes (which has been published) and the English is sometimes a bit awkward.

RESPONSE: We sincerely thank the reviewer for the encouraging and insightful comments. We have added the sugars in the footnotes. As for the language problem, our manuscript has undergone extensive English revisions by an English editing service.